# Clonal haematopoiesis harbouring AML-associated mutations is ubiquitous in healthy adults

Andrew L. Young[1,2], Grant A. Challen[3], Brenda M. Birmann[4] & Todd E. Druley[1,2]

Clonal haematopoiesis is thought to be a rare condition that increases in frequency with age and predisposes individuals to haematological malignancy. Recent studies, utilizing next-generation sequencing (NGS), observed haematopoietic clones in 10% of 70-year olds and rarely in younger individuals. However, these studies could only detect common haematopoietic clones—>0.02 variant allele fraction (VAF)—due to the error rate of NGS. To identify and characterize clonal mutations below this threshold, here we develop methods for targeted error-corrected sequencing, which enable the accurate detection of clonal mutations as rare as 0.0003 VAF. We apply these methods to study serially banked peripheral blood samples from healthy 50–60-year-old participants in the Nurses' Health Study. We observe clonal haematopoiesis, frequently harbouring mutations in *DNMT3A* and *TET2*, in 95% of individuals studied. These clonal mutations are often stable longitudinally and present in multiple haematopoietic compartments, suggesting a long-lived haematopoietic stem and progenitor cell of origin.

[1] Department of Pediatrics, Division of Hematology and Oncology, Washington University School of Medicine, Saint Louis, Missouri 63108, USA. [2] Center for Genome Sciences and Systems Biology, Washington University School of Medicine, Saint Louis, Missouri 63108, USA. [3] Department of Internal Medicine, Division of Oncology, Washington University School of Medicine, Saint Louis, Missouri 63108, USA. [4] Department of Medicine, Channing Division of Network Medicine, Department of Medicine, Brigham and Women's Hospital and Harvard Medical School, Boston, Massachusetts 02115, USA. Correspondence and requests for materials should be addressed to T.E.D. (email: druley_t@wustl.edu).

The advent of cost-effective, next-generation sequencing (NGS) has permitted in-depth analysis of the spectrum of somatic mutations driving clonal evolution in malignancy[1–3]. Subsequently, benign clonal haematopoiesis was identified in healthy individuals[4–7]. Recent studies revealed that malignant and benign haematopoietic clones frequently harbour mutations in the epigenetic modifiers *DNMT3A* and *TET2* (refs 1,8–11). Benign clones were rarely detected before 60 years old, but were detected in 10–20% of individuals older than 70 years old[8–11]. While compelling, these previous studies could only detect common clonal mutations—greater than 0.02 variant allele fraction (VAF)—due to the NGS error rate. Haematopoietic clones detected above this 0.02 VAF threshold have been termed clonal haematopoiesis of indeterminate potential (CHIP) and are associated with an increased risk of the developing haematological malignancy[12].

Recently, the development of error-corrected sequencing (ECS) using single molecule tagging with unique molecular identifiers has permitted the detection of rare variants below the error rate of NGS[13–18]. Here we combined ECS with targeted capture for 54 genes, recurrently mutated in acute myeloid leukaemia (AML) to enable the detection of clonal mutations at VAFs two orders of magnitude lower than the detection limit of NGS. Using these methods, we sought to thoroughly describe the prevalence and mutation profile of rare haematopoietic clones in healthy individuals, determine if these clones are stable longitudinally, and determine if clonal mutations arise in long-lived haematopoietic stem and progenitor cells (HSPCs) or in more committed progenitors. We studied clonal haematopoiesis in longitudinally banked blood samples from middle-aged healthy participants in the Nurses' Health Study (NHS). We found clonal haematopoiesis, predominantly harbouring mutations in *DNMT3A* and *TET2*, in 95% of individuals studied. Many clonal mutations were stable longitudinally and detected in both myeloid and lymphoid lineages, suggesting they arose in long-lived HSPCs.

## Results

**Samples**. We obtained paired buffy coat blood samples, banked ~10 years apart, from 20 healthy participants in the NHS (Methods; Table 1)—a cohort of 121,701 female registered nurses longitudinally studied since 1976 (refs 19–21). The median ages at sample collection were 56.6 and 68.1 years old. This facilitated the investigation of benign clonal haematopoiesis in younger individuals, previously thought to only rarely harbour haematopoietic clones[8–12]. To identify haematopoietic clones, we combined ECS with targeted capture for 568 amplicons in 54 genes frequently mutated in AML (Methods)[14–17]. This enabled us to sequence a tractable subset of the genome, while still querying loci associated with clonal haematopoiesis and AML. Samples were prepared and sequenced in duplicate. We generated an average of 47.7 million paired-end sequencing reads, which yielded an average of 3.4 million error-corrected consensus sequences (ECCSs), per library (Supplementary Table 1).

**Error-corrected NGS**. We modelled position-specific errors in the ECCSs using binomial statistics to identify clonal mutations (Methods). We identified 109 clonal single nucleotide variants (SNVs) in at least one time point <0.2 VAF in 95% (19/20) of individuals. We detected 1–17, mostly exonic, SNVs per individual at 0.0003–0.1451 (median 0.0024) VAF (Fig. 1a; Supplementary Table 2). Of note, the median VAF we observed was 10-fold less than the detection limit governing previous studies of clonal haematopoiesis[8–10]. Separately, we identified nine clonal insertion/deletion variants (indels) in six individuals

(Supplementary Table 3). Indels were identified by ECCS coverage alone, as indel errors were not appropriately modelled by the same statistical framework implemented to identify SNVs.

We were initially concerned that most of the identified rare variants were artefacts introduced during library preparation or sequencing. We first determined that SNV calls were not biased by coverage per amplicon (Supplementary Fig. 1) or by the number of bases captured per gene (Supplementary Fig. 2). Next, we validated these findings using droplet digital PCR (ddPCR)—an orthogonal non-sequencing-based technique for VAF quantification. We designed ddPCR assays for 21 SNVs that had been previously observed in malignancy[22] and for one indel (Methods; Supplementary Fig. 3). The VAFs measured by ECS and ddPCR were highly correlated ($R^2 = 0.98$; Supplementary Fig. 4; Supplementary Table 4), consistent with the previously observed accuracy of ECS[17].

We next compared the mutation profile observed in these rare haematopoietic clones to previous findings in CHIP and AML. We detected 88 exonic clonal SNVs with 58 missense SNVs, 17 nonsense SNVs, 9 synonymous SNVs, 3 splicing SNVs and 1 SNV in a 3′-UTR (Fig. 1b). While exonic variants were detected in only 18 of the 54 genes in the panel, 64% (56/88) occurred in the epigenetic regulators *DNMT3A* and *TET2* (Fig. 1c). We frequently detected multiple *DNMT3A* and *TET2* mutations in the same individual, which were not necessarily in the same clone (Supplementary Fig. 5). The *DNMT3A* SNVs were predominantly nonsense mutations in the 5′ end of the gene or missense mutations in the three functional domains (Supplementary Fig. 6). For comparison, *TET2* SNVs were primarily missense mutations in the functional domains or nonsense mutations throughout the gene (Supplementary Fig. 7), consistent with previous observations of AML[23]. While less prevalent, intronic clonal SNVs were observed in 12 genes with 29% (6/21) detected in *DNMT3A* and 5% (1/21) detected in *TET2* (Supplementary Figs 8,9). The most common type of exonic substitution was the cytosine to thymine (C to T) transition (Fig. 1d), as previously observed in CHIP[8–10]. Conversely, intronic SNVs were evenly distributed across substitution types.

**Longitudinal analyses**. We characterized the temporal stability of these clones by tracking clonal mutations longitudinally within

### Table 1 | Age at sample collection for each NHS participant.

| Participant ID | Collection 1 age (years) | Collection 2 age (years) |
|---|---|---|
| 1 | 53.5 | 64.6 |
| 2 | 51.2 | 63.0 |
| 3 | 52.3 | 64.4 |
| 4 | 53.4 | 64.2 |
| 5 | 52.2 | 64.4 |
| 6 | 57.9 | 69.2 |
| 7 | 60.1 | 71.4 |
| 8 | 56.5 | 68.5 |
| 9 | 58.0 | 69.0 |
| 10 | 54.7 | 66.9 |
| 11 | 63.5 | 74.5 |
| 12 | 56.4 | 67.3 |
| 13 | 56.6 | 68.5 |
| 14 | 60.1 | 71.8 |
| 15 | 57.6 | 67.7 |
| 16 | 54.1 | 65.4 |
| 17 | 51.7 | 63.1 |
| 18 | 65.1 | 76.2 |
| 19 | 64.0 | 75.1 |
| 20 | 62.8 | 74.6 |

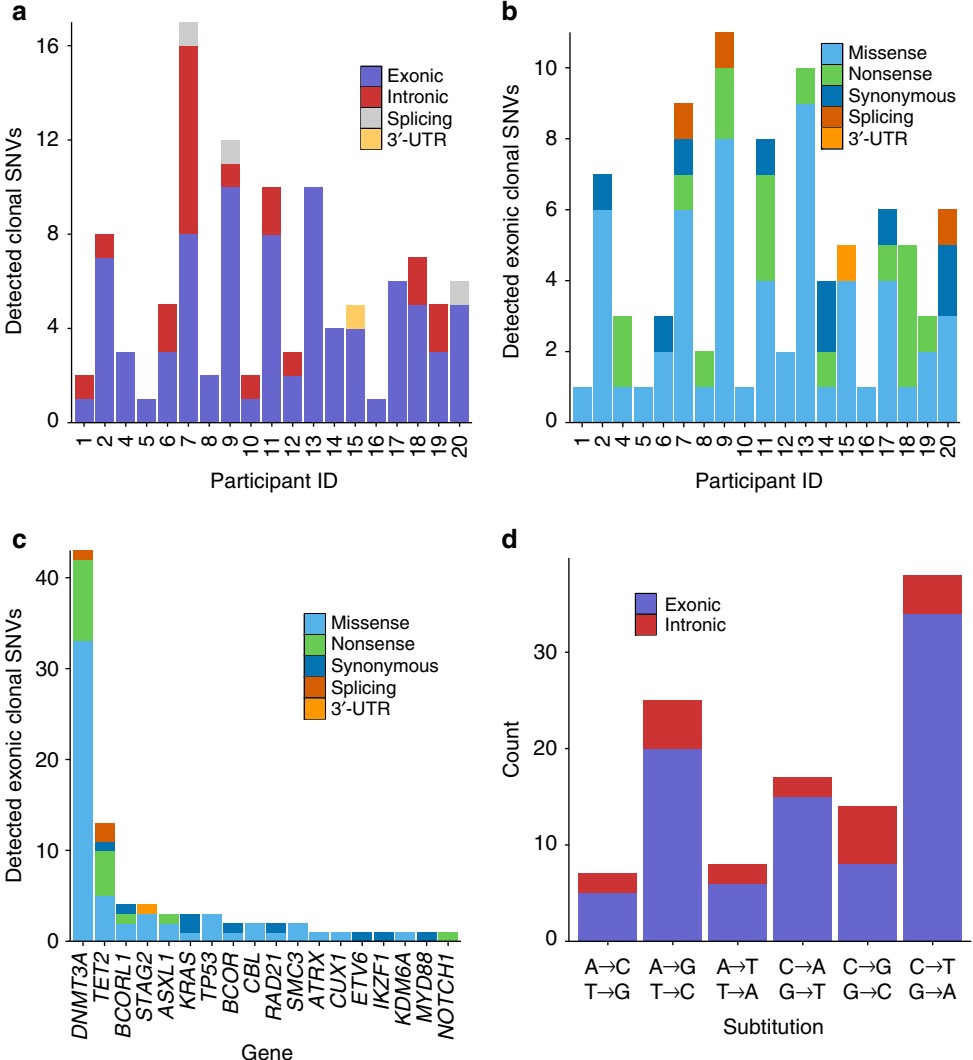

**Figure 1 | Number and characteristics of clonal SNVs detected by ECS in the peripheral blood of healthy adult nurses.** (**a**) Clonal SNVs detected in each individual, colour-coded by annotation. (**b**) Exonic clonal SNVs detected in each individual, colour-coded by predicted effect. (**c**) Detected exonic clonal SNVs organized by gene, colour-coded by predicted effect. (**d**) Distribution of substitution types observed in clonal SNVs.

our 20 study participants. Variants were called independently from paired samples banked ∼10 years apart (Fig. 2). Of the 109 clonal SNVs identified, 27.5% (30/109) were detected at both time points, 13.8% (15/109) were detected at only the first time point and 58.7% (64/109) were detected at only the second time point (Supplementary Table 2). The stability of VAFs observed here was consistent with the previous observations at higher VAFs in a few instances of CHIP[8]. The presence of the same clonal mutations longitudinally suggested that these mutations arose in long-lived HSPCs or committed progenitors.

To further elucidate the cell of origin for clonal haematopoiesis, we sorted 26 samples from 13 individuals into B lymphocyte (CD45 + CD33–CD19 + ), T lymphocyte (CD45 + CD33–CD3 + ) and myeloid (CD45 + CD33 + ) compartments using flow cytometry (Methods; Supplementary Fig. 10). While cell recovery was variable per sample, we observed the same clonal SNVs in both myeloid and lymphoid compartments in 10/13 individuals (Fig. 3; Supplementary Table 5). Frequently, the VAF measured in the bulk sample was approximately equal to the VAF measured in each compartment. These observations were unlikely to have arisen due to contamination, given that variants were often detected at similar VAFs in different sorted compartments.

## Discussion

These findings suggest that clonal haematopoiesis-harbouring mutations in AML-associated genes is nearly ubiquitous (95%) in 50–70-year olds—an age group in which previous studies identified haematopoietic clones in only 5% of individuals[8–11]. Clonal mutations were detected in both the myeloid and lymphoid compartments in samples banked a decade apart in these healthy individuals, clearly demonstrating that these mutations arose in long-lived HSPCs. However, these clonal mutations conferred only a modest proliferation advantage, as most clonal mutations were rare (median 0.0024 VAF) and stable longitudinally. One possible explanation for these observations was that these mutations, often in epigenetic regulators, augmented self-renewal capacity without a concomitant increase in proliferation. This hypothesis may also explain why HSPC number increases and quiescence decreases as a function of age[24,25]. As HSPCs gradually senesce throughout life, the acquisition of these mutations may allow benign clonal haematopoiesis to maintain ostensibly normal blood production years after it would otherwise decline[26]. This hypothesis is supported by work in mice demonstrating that *DNMT3A* loss-of-function mutations in haematopoietic stem cells (HSCs) are associated with an increase in HSC self-renewal without

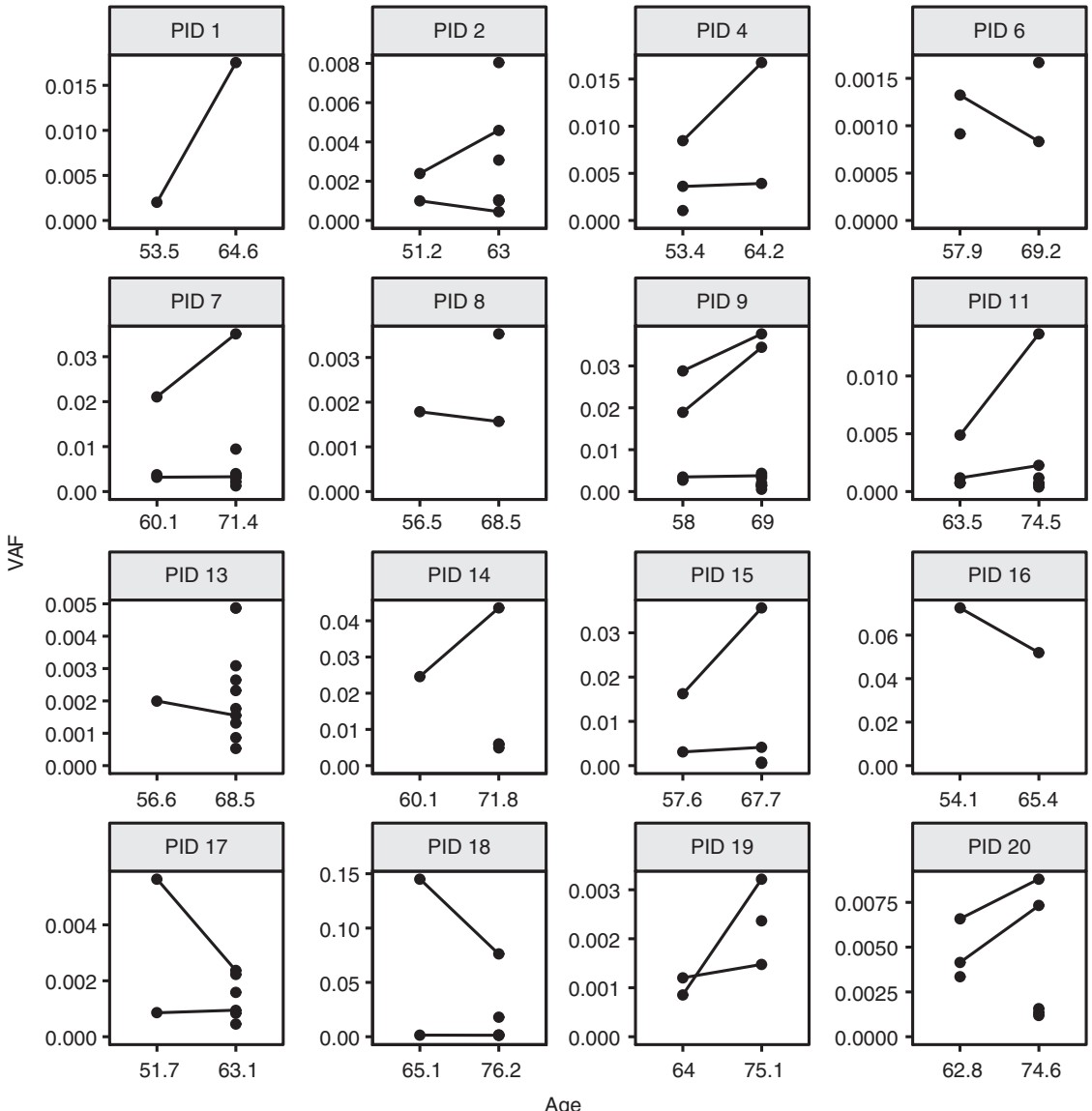

**Figure 2 | Longitudinal detection of clonal SNVs in NHS participants.** Clonal SNVs were detected by ECS in both time points for 16/20 NHS participants. For each participant ID (PID), the VAF measured by ECS was plotted relative to the age at sample collection. Variants detected in both time points were connected with a line.

increasing proliferation[27]. Comparably, *TET2* loss-of-function mutations in mice increase HSC self-renewal and proliferation[28].

While *DNMT3A* mutations are frequently observed in CHIP and AML[1,8–11], we observed a different distribution of *DNMT3A* mutations, specifically at the arginine 882 (R882) residue. Previous studies showed that mutations in *DNMT3A* R882 comprised approximately two-thirds of total *DNMT3A* mutations in AML[29] and one-third of *DNMT3A* mutations in CHIP[8,9]. However, we observed only a single *DNMT3A* R882H variant. These findings suggest that *DNMT3A* R882 mutations potently drive clonal expansion, explaining their prior detection in common CHIP clones (median 0.11 VAF)[8] and their rarity in these lower frequency clones.

The detection limit of ECS was ∼ 1:10,000 cells. Thus, given an estimated 11,000 HSCs in adults, of which only a fraction actively contribute to haematopoiesis at any given time[30], we expected to observe unique somatic mutations marking each active HSC (a random distribution of synonymous and nonsynonymous mutations throughout the 54 AML-associated genes captured). Instead, over half of the detected mutations were in *DNMT3A* or

*TET2*. This observation alone could have occurred if *DNMT3A* and *TET2* were hotspots of somatic mutation. However, 89% (78/88) of the detected exonic mutations were nonsynonymous, truncating or splicing mutations. Given this skew towards presumed functional mutations, it was more likely that these haematopoietic clones were enriched by selection.

Due to technical limitations of our methods, we likely under-reported the number of clonal mutations present in each individual. Specifically, we likely underreported the number of C to T (G to A) substitutions present in these rare haematopoietic clones due to the stringency of the binomial variant calling strategy and the background rate of cytosine deamination, which is a predominant artefact of ECS[14,16,31]. Here 38/109 substitutions identified were C to T (G to A) substitutions. Conversely, in previous studies of CHIP and AML, C to T (G to A) substitutions comprised ∼ 50–60% of identified substitutions[8,9,32]. In addition, the binomial statistical framework underreported hotspot mutations occurring in multiple individuals. However, in our raw data we only observed a single likely instance of an uncalled hotspot mutation—a *DNMT3A* R882H variant in individual 5

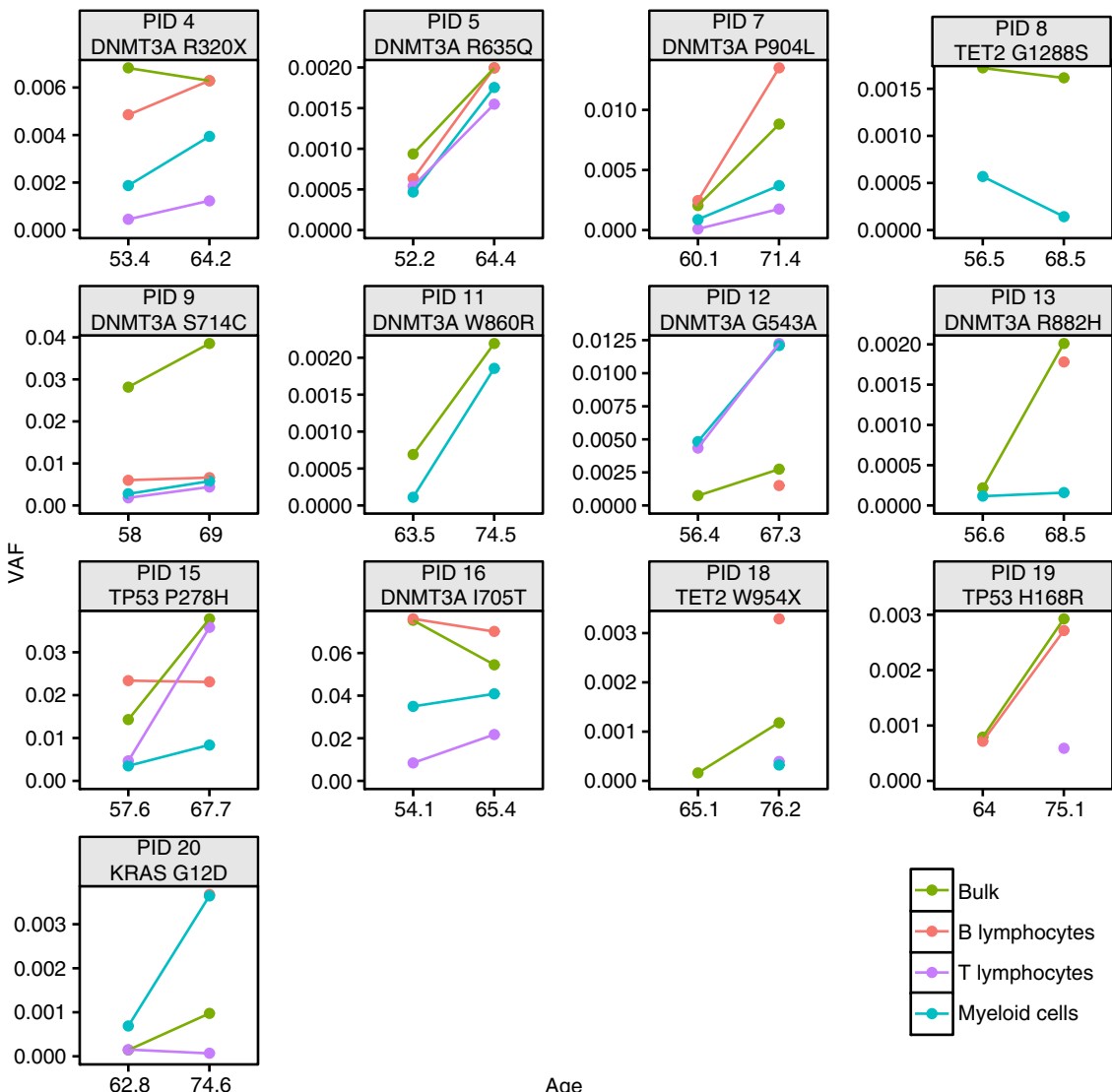

**Figure 3 | Haematopoietic compartment-specific detection of clonal SNVs in NHS participants.** Paired buffy coat samples from 13 individuals were sorted into B lymphocyte (pink), T lymphocyte (purple) and myeloid (blue) compartments using flow cytometry. For each NHS participant ID (PID), a single SNV, detected by ECS, was selected for compartment-specific quantification by ddPCR. Variants detected in both time points were connected with a line. The VAF measured by ddPCR in the bulk sample (green) was included for comparison.

observed at a lower VAF than the variant reported in individual 13. In addition, we could not routinely co-localize mutations within the same haematopoietic clone. However, we co-localized mutations in three instances where they co-occurred in the same sequenced reads (participant ID (PID) 2, *TET2* R1216G/A1217A; PID 13, *DNMT3A* G498V/C494F; PID 14, *KRAS* A66A/S65S). Future adaptations of this technology could address these limitations by targeting a larger capture panel and implementing single-cell sequencing approaches.

In summary, we demonstrate that clonal haematopoiesis, originating in long-lived HSPCs, is far more common than previously thought in healthy middle-aged adults. Despite its prevalence, clonal haematopoiesis shares many mutations with AML, raising additional questions regarding the sequence of mutation acquisition and cooperating events necessary for malignant transformation. Furthermore, in previous studies of CHIP the detection of a haematopoietic clone (at any age) was associated with an 11–13-fold increased risk of developing a haematological malignancy[8,9]. These earlier findings suggested that CHIP was comparable to monoclonal gammopathy of undetermined significance and monoclonal B-cell lymphocytosis, which are benign clonal proliferative conditions that occasionally progress to haematological malignancy[6,7,12]. Conversely, our findings suggest that clinically silent clonal haematopoiesis is present in almost all individuals by middle age, and that progression to haematological malignancy is exceptionally rare. Given the current public interest in precision medicine[33], these findings have practical implications for sequencing-based screening of nascent malignancy or recurrence. Future research must focus on reliably distinguishing benign clonal haematopoiesis, however rare, from malignant clonal haematopoiesis that could drive transformation and relapse. This imperative extends to sequencing-based non-invasive screening[34], which will require even finer discrimination between nascent malignancy and benign clonal expansion.

## Methods

**Study population.** The NHS began in 1976 with 121,701 female United States registered nurses age 30–55 years old who returned an enrolment questionnaire, which queried medical history, anthropometric measures and lifestyle/environmental risk factors[19]. Since enrolment, the participants have returned biennial follow-up

questionnaires to update information on potential exposures and diagnoses of chronic disease. To date, follow-up rates have been consistently high (>90%). In 1989–1990, 32,826 women provided a heparinized whole blood sample by methods described previously[20]. In 2000–2001, 18,743 of the women who had provided a sample in 1989–1990 provided a second whole blood sample using the same protocol[21]. In brief, participants willing to provide blood samples received kits that included all supplies necessary for their collection and overnight return (including a chill pack), and a brief questionnaire. Upon receipt, specimens were separated into plasma, buffy coat and red blood cell fractions, and frozen in liquid nitrogen. Informed consent to participate in the NHS was implied by return of the enrolment and follow-up questionnaires; written informed consent was obtained for biomarker studies at time of blood collection.

Among women who provided blood samples in 1989–1990 and 2000–2001, we identified 20 with no history of cancer or other major chronic disease. De-identified aliquots from those 40 buffy coat samples were prepared and shipped overnight to Washington University for the detection of persistent rare haematopoietic clones harbouring AML-associated somatic mutations as described below. Since each sample was de-identified and the capture space for targeted genomic DNA sequencing was not enough to enable the individual identification (141 kb per person), the Washington University Human Research Protection Office deemed this study as non-human research.

**Sample preparation for ECS.** Genomic DNA was extracted from 50 µl of purified buffy coat from each sample using the QIAmp DNeasy Blood and Tissue Kit (Qiagen) with MinElute columns (Qiagen) instead of standard columns to facilitate elution in a lower volume (three 30 µl elutions). The concentration of the extracted genomic DNA was measured using the Qubit dsDNA HS Assay Kit (Life Technologies). Enrichment of 568 amplicons in 54 genes (141 kb) commonly mutated in AML was performed using 250 ng of genomic DNA via the Illumina TruSight Myeloid Panel (Illumina). Technical replicates were prepared for each sample (80 libraries total). Following extension–ligation, the amplified fragments were eluted in 50 mM NaOH. Recovered fragments were amplified using the Q5 High-Fidelity 2x Master Mix (New England Biolabs) in a 75 µl reaction (37.5 µl 2x master mix, 20 µl DNA in 50 mM NaOH, 2 µl Tris-HCl pH 7.5 and 0.4 µM i5/i7 primers). Illumina's standard i7 primers were used to enable sample multiplexing. The i5 primer was redesigned to contain a random 16 nucleotide index to facilitate ECS (Supplementary Table 6). The following conditions were used for amplification: 98C for 30 s; six cycles of 98 °C for 10 s, 66 °C for 30 s, 72 °C for 30 s; 72 °C for 2 min; hold 10 °C. The PCR reaction was purified using a modified Ampure bead (Beckman Coulter) clean up to purify the amplified fragments (>400 bp). A modified poly-ethylene glycol (PEG) solution was made containing 382.5 µl 50% wt/vol PEG (Sigma), and 562.5 µl 5 M NaCl and 555 µl ddH$_2$O. One-hundred microlitres of beads were washed twice with ddH$_2$O to remove the stock PEG solution. One-hundred-fifty microlitres of the modified PEG solution was added to the washed Ampure beads with the 75 µl PCR reaction and otherwise purified using the standard Ampure protocol. The fragments were eluted in 20 µl ddH$_2$O and the concentration of each library was quantified with Qubit (Life Technologies).

**Quantification by ddPCR.** Our goal was to generate each ECS library from 4M uniquely tagged molecules. We quantified each library's concentration using the QX200 ddPCR platform (Bio-Rad). A 2 µl aliquot of each library was diluted 1,000-fold and quantified in duplicate wells. Each well contained the following reaction mixture: 10 µl 2× EvaGreen 2× ddPCR master mix (Bio-Rad), 5 µl 1:1,000 diluted ECS library, 100 nM P5/P7 primers (Supplementary Table 6) and ddH$_2$O added to 20 µl total. Droplets were generated using the standard Bio-Rad protocol. Amplification was completed using the following conditions: 95 °C for 5 min; 40 cycles of 95 °C for 30 s, 66 °C for 1 min; 4 °C for 5 min; 90 °C for 5 min; 4 °C hold. With the calculated concentration, we aliquotted the appropriate volume of each library to introduce 4M molecules into the subsequent amplification step.

**Amplification and normalization.** Following ddPCR quantification, 4M molecules for each library were amplified using Q5 High-Fidelity 2× Master Mix (New England Biolabs) using 1 µM P5/P7 primers (Supplementary Table 6) in a 50 µl reaction under the following conditions: 98 °C for 30 s; 16 cycles of 98 °C for 10 s, 66 °C for 30 s, 72 °C for 30 s; 72 °C for 2 min; 4 °C hold. The PCR reaction was purified using the modified Ampure bead clean up. One-hundred microlitres of beads were washed twice with ddH$_2$O and replaced with 100 µl of the modified PEG solution described above. The PCR reaction was then added to the mixture and purified using the standard protocol. The fragments were eluted in 20 µl ddH$_2$O. A 2 µl aliquot of each library was diluted 10-fold and quantified on the Agilent 2200 Tape Station. Libraries were then pooled in equimolar groups of eight. Once pooled, each library was again quantified on the Tape Station and submitted for sequencing.

**Sequencing.** Each library was sequenced on the Illumina NextSeq platform using a 300 cycle high output kit as specified by the manufacturer. Approximately 5–10% of PhiX control DNA was spiked into each sequencing experiment. Each sequencing run contained roughly 400 M paired-end 144 bp reads with

corresponding 16 bp unique molecular index and 8 bp sample-specific index sequences. Sequenced reads were demultiplexed by sample-specific index allowing for at most one mismatch in the index sequence (Supplementary Table 1). Raw sequence reads were aligned to the PhiX genome using Bowtie 2 (ref. 35). Sequence reads that did not align to PhiX were retained for the subsequent analysis (below).

**ECS analysis.** The first 30 nucleotides of each sequenced read were hard clipped to remove the primer sequences from the TruSight Myeloid panel. Next, the sequenced read pairs tagged with the same random index sequence (originating from the same uniquely tagged DNA molecule) were aligned to each other to generate read families in a manner similar to the previously published methods[14–17]. Read families were required to have five or more reads sharing the same index sequence. Paired-end reads within a read family were error corrected to generate an ECCS in a stepwise manner. First, at every position, the nucleotides called by each sequence read were compared and a consensus nucleotide was called if there was at least 90% agreement between the reads. If there was <90% agreement, an N was called in the consensus sequence at that position. Errors that occurred during the library preparation and sequencing were corrected or removed because they were not shared between different reads within a read family. Second, an ECCS was discarded if >10% of the 228 nucleotides comprising the paired-end read were reported as N nucleotides. ECCSs were then locally aligned to UCSC hg19/GRCh37 using Bowtie2 and realigned with GATK's Indel Realigner[36]. Next, the aligned ECCSs were processed with Mpileup using the parameters -BQ0 -d 10,000,000,000,000. This removed the coverage thresholds to ensure that all of the pile up output was returned regardless of VAF or coverage. The parsed pile up output was further filtered to ignore positions with <1,000x ECCS coverage or outside of the Illumina TruSight Myeloid target space. In addition, germline variants identified by the 1,000 Genomes Project above a 0.01 minor allele fraction were excluded from the analysis.

We implemented a position-specific binomial error model to improve rare clonal SNVs calling as described previously[17]. For each sample, we generated a nucleotide position-specific error profile using all sequenced libraries that were not from the same individual. A variant was called if: (a) the binomial $P$ value was <0.05 after Bonferroni correction; (b) the variant was observed in at least five ECCSs; (c) the VAF was >0.0001; and (d) the variant was identified with criteria a–c in at least two replicates from one of the two time points. Likely, clonal SNVs (<0.2 VAF) were reported and annotated using Annovar[37], with the COSMIC 68 (ref. 22) and 1,000 Genomes (October 2014 release)[38] databases. The amino-acid substitutions were predicted based on the canonical transcript reported in the GENCODE (v22)[39] as retrieved from the UCSC Table Browser[40].

We identified potential insertion/deletion (indel) events using VarScan 2 (ref. 41), from the filtered Mpileup output (described above), with the following parameters: min-coverage 1,000; min-reads2 5; min-var-freq 0.001; strand-filter 0; and output-vcf 1. We filtered out single-nucleotide indels in homopolymer runs at least four nucleotides long and indels that were observed in multiple samples to remove technical artefacts in the variant calling. We reported likely clonal indels (<0.2 VAF) that were detected in at least two replicates from one of the collection time points. Reported indels were annotated with Annovar[37] as described previously.

**ddPCR validation.** We validated 21 SNVs and 1 indel using the ddPCR probe assay (Bio-Rad)[42]. Probes were designed by Bio-Rad based on MIQE guidelines for the quantitative digital PCR[43]. All reagents were purchased from Bio-Rad. For each sample and control, 45 ng of the genomic DNA was aliquoted per well of generated droplets. We generated between 8 and 32 wells of droplets for each validation experiment, depending on the expected VAF for the assayed mutation. Each control sample was assayed with the same number of wells as the corresponding sample. Droplets were generated on the QX200 Droplet Generator (Bio-Rad) and assayed on the QX200 droplet reader (Bio-Rad) using the standard protocol[42]. The VAF was estimated from droplets lacking the reference allele and the Poisson-estimated number of singleton droplets as described previously[17].

**Flow cytometry.** Cells were sorted from the buffy coat samples using a Sony iCyt Synergy SY3200 BSC 17-colour, 5-laser cell sorter (Sony Biotechnology Inc.) and analysed with FlowJo (Treestar) using standard protocols (Supplementary Fig. 10). Cells were stained with the following antibodies (BioLegend): CD45 (BV-421), CD33 (APC), CD19 (FITC) and CD3 (PE-CY7) per the manufacturer's instructions. Variants were detected in purified cell populations using the ddPCR assay described previously.

**Data availability.** The sequencing data have been deposited into the NCBI Sequence Read Archive under accession number SRP078948. All other relevant data are included in the article or supplementary files, or available from the authors upon request.

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

## Acknowledgements

We thank the participants and staff of the Nurses' Health Study for their valuable contributions, D. Link for discussions and feedback on the manuscript and A. Kothari for assistance with the cell-sorting experiments. The authors assume full responsibility for analyses and interpretation of these data. Funding for this project was provided by the National Institutes of Health (UM1 CA186107, R01 CA49449 and R01 CA149445), the Children's Discovery Institute of Washington University and St Louis Children's Hospital (MC-II-2015-461), and Hyundai Hope on Wheels (2015Q3-3).

## Author contributions

A.L.Y. designed and performed the research, analysed the data and wrote the manuscript. G.A.C. contributed to the flow cytometry assay. B.M.B. provided samples from the Nurses' Health Study and contributed to the study design. T.E.D. supervised all of the research and edited the manuscript, which was approved by all co-authors.

## Additional information

**Competing financial interests:** The authors declare no competing financial interests.

