## [Peer Review File · Nature Communications]

Reviewer #1 (Remarks to the Author): Expert in haematopoiesis

In this manuscript, Young and colleagues use focused, extremely deep sequencing to interrogate a cohort of 20 presumably healthy individuals randomly selected from the Nurses' Health Study for the presence of clonal hematopoiesis (as defined by the presence of pathogenic mutations found recurrently in AML and other myeloid neoplasms). They show that low-level clonal hematopoiesis is nearly ubiquitous, is highly enriched for inactivating mutations in DNMT3A and TET2, is present at the level of the hematopoietic stem cell, and in most cases remains stable over a 10-year period. The work is carefully conducted and well validated, and represents an important contribution to our understanding of clonal hematopoiesis and its implications.

1. One concern I whether a significant percentage of variants detected are artifacts. The authors identify 109 SNVs, of which 71 are non-C->T transitions (65.1%). I compared this to three other data sets: Welch et al Cell 2012 identified 37/82 mutations in normal HSCs to be non-CT (45.1%, $p=0.008$ by Fisher exact test), AML TCGA data identified 762/2008 mutations to be non-CT (38.0%, $p=3.0 \times 10^{-8}$), Jaiswal et al NEJM 2014 identified 267/609 mutations to be non-CT (43.8%, $p=4.4 \times 10^{-5}$). So in these three datasets, the proportion of non-CT mutations is ~40-45%, whereas it is substantially higher in the submitted manuscript, and the p-values indicate that this difference is virtually impossible to be due to chance.

These results have 2 possible explanations. The first is that the method in the submitted manuscript substantially underestimates CT mutations. This seems unlikely. The second possibility is that the method in the submitted manuscript substantially overestimated the number on non-CT mutations, and that a large proportion of these (as many as 50%) could be artifacts. The authors should look for potential causes of artifacts in their data set, including things like oxo-G filtering. At a minimum, they should point out that their data substantially differs from exome sequencing data previously published, and that this difference cannot be due to chance alone. The best option would be for the authors to perform a negative control on cells that should have close to zero chance of having clonal hematopoiesis, such as cord blood HSCs.

2. The authors should reword some of their conclusions. The authors make the statement: "Conversely, our findings suggest that CHIP is present in almost all individuals by middle age, and that progression to hematopoietic malignancy is exceptionally rare." CHIP was defined as a mutant allele fraction greater than 2%. The authors need to be clear that the rare mutations identified here do not meet the definition of CHIP as currently defined.

There is a clear relationship between size of clone and risk of hematologic malignancy that has been reported, for example in Jaiswal et al NEJM 2014. The expected finding would be that the rare clones do not have a significant impact, but the authors come close to saying that that the clinical impact of the larger clones identified in multiple other studies are inconsequential. The authors need to be careful not to conflate the small clones found here, whose clinical consequences we do not know, with those described in the original CHIP papers.

4. The authors make the statement: "These findings have profound implications for precision medicine, and specifically for sequencing-based monitoring of response to treatment and recurrence." The authors should be careful here. The authors may be referring to finding DNMT3A and other clones in samples taken from AML and other patients who have achieved morphologic remission. However, the observation in this paper that low-level clones are ubiquitous in native hematopoiesis among healthy individuals should not be used to back assumptions about behavior in response to selective pressures (e.g. chemotherapy), which has not yet been fully explored. In fact, we know that patients who do not clear mutations after chemo have an increased risk of relapse and death (Klco et al, JAMA 2015).

5. Regarding the finding that clones are stable over time, the authors raise one hypothesis, that the mutations alter self-renewal without altering proliferation. This is one of many hypotheses, and

the effects of these mutations has been studied in detail in murine models. The authors should note that there are other hypotheses and perhaps cite the critical papers in this field.

6. In Figures 1a and 1b, the authors could show a bar for the expected number of mutations of each type that would be expected in the particular capture set used based on the number of nucleotides of each type and perhaps the background mutational processes.

7. The use of the term "subclonal" is confusing. Typically the term "subclone" is used in reference to a known parental clone, whereas here it seems to be used more in the sense that the term "clonal" (that is, all progeny of a single mutated HSC) is used. Is there a particular reason for using the term "subclonal" in this context?

8. Reference 15 is outdated. The correct reference is

Wong, T.N. et al. Role of TP53 mutations in the origin and evolution of therapy-related acute myeloid leukemia. *Nature* 518, 552-5 (2015).

Reviewer #2 (Remarks to the Author): Expert in haematopoiesis and AML

This is a very well done and interesting paper.

1. One question would be whether any of the rare subclones can functionally enrich; can the authors take cells from a patient they sorted, and see if any clones increase in frequency with colony assays? In vivo studies would be of even more value but are clearly outside scope.

Reviewer #3 (Remarks to the Author): Expert in haematopoiesis and sequencing

This manuscript describes sensitive detection of clonal hematopoiesis in paired blood specimens, banked ~10 years apart, from 20 healthy individuals aged medians of 56.6 and 68.1 years old, using barcoded targeted capture sequencing of 54 AML-associated genes. The most significant finding here is very high frequency of clonal hematopoiesis, which tended to be stable over years and nevertheless, increase with the second samplings; 19 of 20 had subclones carrying 1-17 somatic mutations, which frequently involved epigenetic regulators, such as DNMT3A and TET2. At the second time point, 13.8% of the mutations found at the first time point were lost, whereas 27.5% persisted and 58.7% were newly acquired. The results are of potential interest, but there are several concerns to be addressed.

Major points:

- 1) The significance of the present study critically depends on the accuracy of detection of mutations having very low allele frequencies. Authors evaluated the accuracy by seeing distribution of mutations in terms of coverage per amplicon and the number of mutations per gene, as well as ddPCR for selected mutations. My concern is that ddPCR validation is biased for mutations having high prior probability, such as those in DNMT3A and TET2. How is the accuracy of mutations in other less frequently mutated genes having low allele frequency?
- 2) There was discordance in the results of the duplicated experiments for several mutations within the list of 109 mutations (Supplementary Table 2); some mutations were not detected or showed substantially different VAFs in duplicated samples. These mutations should not be considered as true positive if ever the duplicated experiments are meaningful. Otherwise, all such samples should be intensively validated by ddPCR or other independent experiments.
- 3) In Supplementary Figure 4, which of the results from duplicated experiments was used for the plots, or authors used mean values? Again, some results in duplicated experiments were very different. Also it does not seem to be appropriate to analyze very different VAFs (ranging from

0.0003-0.1451 (~500 times!) in the manner presented. The plot should be in log scale and statistical method should be improved.

4) Authors are requested to test whether or not newly detected mutations in the second time point was present from since the first time point using ddPCR.

5) Were there any other mutations showing >0.2 of VAFs?

6) Authors speculated that multiple DNMT3A and TET2 mutations may not resided in the same clone. It should be tested whether or not these mutations were found in the same sequencing reads for those mutations located in the vicinity, if ever present.

7) Authors generated a nucleotide position-specific error profile for each sample, using all sequenced libraries that were not from the same individual. But this would lead to elimination of hotspot mutations that may be present recurrently in different individuals, such as RAS mutations. There was low frequency of R822 DNMT3A mutations and other hot spot mutations.

8) Many intronic SNVs were found in DNMT3A and other genes (Supplementary Figure 8), which is also of interest. Given that these mutations are likely non-functioning, what was the status of mutations in other AML-related genes to explain what mutations or genetic events drove the clonal evolution of cell having non-functional mutations. Also were there any specific mutation signatures, such as transcription-coupled repair?

In this manuscript, Young and colleagues use focused, extremely deep sequencing to interrogate a cohort of 20 presumably healthy individuals randomly selected from the Nurses' Health Study for the presence of clonal hematopoiesis (as defined by the presence of pathogenic mutations found recurrently in AML and other myeloid neoplasms). They show that low-level clonal hematopoiesis is nearly ubiquitous, is highly enriched for inactivating mutations in DNMT3A and TET2, is present at the level of the hematopoietic stem cell, and in most cases remains stable over a 10-year period. The work is carefully conducted and well validated, and represents an important contribution to our understanding of clonal hematopoiesis and its implications.

1. One concern I have is whether a significant percentage of variants detected are artifacts. The authors identify 109 SNVs, of which 71 are non-C->T transitions (65.1%). I compared this to three other data sets: Welch et al *Cell* 2012 identified 37/82 mutations in normal HSCs to be non-CT (45.1%, $p=0.008$ by Fisher exact test), AML TCGA data identified 762/2008 mutations to be non-CT (38.0%, $p=3.0 \times 10^{-8}$), Jaiswal et al *NEJM* 2014 identified 267/609 mutations to be non-CT (43.8%, $p=4.4 \times 10^{-5}$). So in these three datasets, the proportion of non-CT mutations is ~40-45%, whereas it is substantially higher in the submitted manuscript, and the p-values indicate that this difference is virtually impossible to be due to chance.

These results have 2 possible explanations. The first is that the method in the submitted manuscript substantially underestimates CT mutations. This seems unlikely. The second possibility is that the method in the submitted manuscript substantially overestimated the number on non-CT mutations, and that a large proportion of these (as many as 50%) could be artifacts. The authors should look for potential causes of artifacts in their data set, including things like oxo-G filtering. At a minimum, they should point out that their data substantially differs from exome sequencing data previously published, and that this difference cannot be due to chance alone. The best option would be for the authors to perform a negative control on cells that should have close to zero chance of having clonal hematopoiesis, such as cord blood HSCs.

Response: This is an excellent point that initially puzzled us as well. After careful analysis of the data and our statistical process, we conclude that the most likely explanation for the deviation from the previously observed non-CT mutation rate is that error-corrected sequencing underestimates CT mutations. First, to clarify, error-corrected sequencing is qualitatively different from "deep sequencing," so our response here is not at all intended to call into question any of the existing work cited by the reviewer. Conventional-NGS sequencing, which the aforementioned studies were based on, has a roughly 1-3% error-rate, which is not mitigated by increased sequencing depth. Here, by using unique molecular identifiers (UMIs) to correct sequencing errors, we directly observe the background rate of cytosine deamination (major source of erroneous C to T calls) and guanine oxidation to 8-oxo guanine (major source of erroneous G to T calls) that occurs *below the rate of sequencing error and would be invisible using standard NGS sequencing approaches*. These artifacts

have been previously reported in papers integrating UMIs for DNA sequencing (Schmitt *PNAS* 2012, Lou *PNAS* 2013). To illustrate this point, plotted below is a cumulative distribution function of aggregate substitutions called for the 80 samples sequenced. This figure demonstrates that, in general, C to T (G to A) substitutions (cytosine deamination) occurs more frequently than other types of substitutions *in the raw data*. Given the higher background rate of C to T substitutions, true clonal mutations present at low frequency would more likely be indistinguishable from cytosine deamination artifacts.

Our concern that this increased rate of detectable C to T substitutions would lead to an increased rate of C to T false positive variant calls prompted us to introduce binomial statistics in order to model the position specific error rate of all possible background substitutions at each base position in all 80 samples. This was possible because we found that the incidence of artifacts was strongly position-specific. Likewise, the frequencies at which these artifacts occurred were largely *consistent between samples* at each position captured by the panel. Thus, the position-specific binomial model acted as a filter to prevent us from calling common artifacts at base positions where most or all of the samples had a small portion of reads with similar artifacts. Likewise, the stringency of this method reduces our sensitivity to identify C to T (and to a lesser extent G to T) substitutions, while improving specificity. To optimize the stringency of our ECS variant calling, we ultimately opted to call variants *independently* from two technical replicates separately prepared for sequencing and *only report variants that were called independently in both replicates*. Given that rare variant calling from error-corrected sequencing data is still experimental, without established pipelines such as GATK, we validated 21 of these called variants with a broad range of VAFs using droplet digital PCR (ddPCR). The validation rate was 100% and the VAF measured by ddPCR was highly concordant with the VAF measured by error-corrected sequencing (see Supplementary Figure 4).

To highlight this difference between the spectrum of detected mutations in our work compared to similar datasets and our reasoning for the difference, we have added the following text to paragraph 2 on page 4 outlining limitations of this technique:

“Specifically, we likely underreport the number of C to T (G to A) substitutions present in these rare hematopoietic clones due to the stringency of the binomial variant calling strategy and the background rate of cytosine deamination, which is a predominant artifact of error-corrected sequencing^{16,18,29}. Here 38/109 substitutions identified were C to T (G to A)

substitutions. Conversely, in previous studies of CHIP and AML, C to T (G to A) substitutions comprise approximately 50-60% of identified substitutions^{9,10,30} .”

2. The authors should reword some of their conclusions. The authors make the statement: "Conversely, our findings suggest that CHIP is present in almost all individuals by middle age, and that progression to hematopoietic malignancy is exceptionally rare." CHIP was defined as a mutant allele fraction greater than 2%. The authors need to be clear that the rare mutations identified here do not meet the definition of CHIP as currently defined.

There is a clear relationship between size of clone and risk of hematologic malignancy that has been reported, for example in Jaiswal et al NEJM 2014. The expected finding would be that the rare clones do not have a significant impact, but the authors come close to saying that that the clinical impact of the larger clones identified in multiple other studies are inconsequential. The authors need to be careful not to conflate the small clones found here, whose clinical consequences we do not know, with those described in the original CHIP papers.

Response: Thank you for bringing this to our attention. The statement (paragraph 3, page 4) has been reworded as to not call the rare hematopoietic clones detected in these healthy individuals CHIP:

“Conversely, our findings suggest that clinically silent clonal hematopoiesis is present in almost all individuals by middle age, and that progression to hematopoietic malignancy is exceptionally rare.”

We acknowledged/cite the previous CHIP studies and the increased risk of hematopoietic malignancy observed with higher frequency clones in the two preceding sentences. Conceptually, clonal hematopoiesis (whether CHIP, CHOP or fulminant malignancy) must exist on a continuum that starts as a very rare event, and the real questions are why so few of these events continue to proliferate and even fewer undergo transformation. We fail to see how our conclusion discounts the clinical impact of larger clones as inconsequential, and that was certainly not our intent. Our primary conclusion is that these rare hematopoietic clones detected by error-corrected sequencing are of unknown/indeterminate clinical potential compared to the higher frequency clones detected in the earlier CHIP reports.

4. The authors make the statement: "These findings have profound implications for precision medicine, and specifically for sequencing-based monitoring of response to treatment and recurrence." The authors should be careful here. The authors may be referring to finding DNMT3A and other clones in samples taken from AML and other patients who have achieved morphologic remission. However, the observation in this paper that low-level clones are ubiquitous in native hematopoiesis among healthy individuals should not be used to back assumptions about behavior in response to selective pressures (e.g. chemotherapy), which has not yet been fully explored. In fact, we know that patients who do not clear mutations after chemo have an increased risk of relapse and death (Klco et al, JAMA 2015).

Response: Thank you for pointing out this potential point of confusion. We agree; these results do not provide any evidence for how these clones will respond to chemotherapy and we do not make any assertions in that regard. It seems quite reasonable that the majority of the mutations identified in our work are of negligible clinical impact. Also, there are 20 years of concrete minimal residual disease data showing that the relative risk of relapse and survival is correlated to the frequency of residual disease following therapy, whether that is quantified by cell surface markers or sequencing assays, and we do not dispute those conclusions. Our goal is to highlight that the normal incidence and distribution of clonal hematopoiesis is far from well described and even less understood. Subsequently, the implementation of sequencing-based assays of peripheral blood samples for cancer-associated mutations in order to screen for nascent malignancy or relapse should be done with caution. We are not advocating this position, just responding to the public dialogue about using non-invasive peripheral blood sequencing as a diagnostic modality.

We have rephrased the statement in paragraph 3 of page 4:

“Given the current public interest in precision medicine³³, these findings have practical implications for sequencing-based screening of nascent malignancy or recurrence. Future

research must focus on reliably distinguishing benign clonal hematopoiesis, however rare, from malignant clonal hematopoiesis that could drive transformation and relapse.”

5. Regarding the finding that clones are stable over time, the authors raise one hypothesis, that the mutations alter self-renewal without altering proliferation. This is one of many hypotheses, and the effects of these mutations has been studied in detail in murine models. The authors should note that there are other hypotheses and perhaps cite the critical papers in this field.

Response: After consultation with all coauthors and local colleagues, we have added the citations describing the *DNMT3A* knockout (Peggy Goodell’s work) and *TET2* knockout mouse HSC experiments (Ross Levine’s work), which are consistent with our interpretation that these mutations increase self-renewal without necessarily increasing proliferation. Our revised text is as follows:

“This hypothesis is supported by work in mice demonstrating that *DNMT3A* loss-of-function mutations in HSCs are associated with an increase in HSC self-renewal without increasing proliferation²⁷. Relatedly, *TET2* loss-of-function mutations in mice increase HSC self-renewal and proliferation²⁸.”

We did not intentionally omit any seminal papers in this space, but have been unable to identify other models supporting other interpretations of these results. Should the reviewer or editor have additional suggestions, we would be more than happy to include mention of these works in our text.

6. In Figures 1a and 1b, the authors could show a bar for the expected number of mutations of each type that would be expected in the particular capture set used based on the number of nucleotides of each type and perhaps the background mutational processes.

Response: We are unaware of any published data describing the profile of clonal hematopoiesis below 3% VAF. At the outset, by assaying variants at frequencies at least two orders of magnitude below existing studies of CHIP, we expected to see the stochastic “noise” of mutations dispersed across the genome. However, that is not what we found. There are no “priors” for us to estimate what we would find across the number of genes and different sequence contexts included in the Illumina TruSight Myeloid Sequencing Panel. We have demonstrated that our sequencing results are not due to A) the number of absolute bases queried for each gene (Supplementary Figure 2) and B) the relative amount of sequencing coverage per amplicon in the panel (Supplementary Figure 1). The previous CHIP studies similarly reported the number of mutations in each gene and the number of mutations per person, so we are unclear what additional bar of information the reviewer would like included in Figure 1.

Similarly, it is beyond the scope of this study to attempt to describe the background mutational process underlying this phenomenon. The fact that we do not observe a uniform distribution of mutations across all genes suggests that mutations in the genes where we do NOT find mutations may be so deleterious that these clones fail to expand or are rapidly eliminated. We do not find “hotspots” in *DNMT3A* and also find that gene has the most called intronic mutations, which is not accounted for by coverage or targeted genomic space. The most obvious reconciliation of these results is that mutations in the 18 genes found to be mutated, particularly *DNMT3A* and *TET2*, impart some beneficial (or at least benign) impact to the cells. Yet, without seeing significant expansion of these clones 10 years later, the benefit to the cells must lie in a parameter other than cellular proliferation. We have not expounded upon this in our text because we have only speculation in this regard, and future studies must focus on the mechanisms driving this mutational skewing.

7. The use of the term “subclonal” is confusing. Typically the term “subclone” is used in reference to a known parental clone, whereas here it seems to be used more in the sense that the term “clonal” (that is, all progeny of a single mutated HSC) is used. Is there a particular reason for using the term “subclonal” in this context?

Response: This is an excellent point and we apologize for any confusion. The text has been edited to refer to identified clonal mutations instead of subclonal mutations.

8. Reference 15 is outdated. The correct reference is

Wong, T.N. et al. Role of *TP53* mutations in the origin and evolution of therapy-related acute myeloid leukemia. *Nature* 518, 552-5 (2015).

Response: Thank you for catching this outdated reference. The citation was corrected.

Reviewer #2 (Remarks to the Author): Expert in haematopoiesis and AML

This is a very well done and interesting paper.

1. One question would be whether any of the rare subclones can functionally enrich; can the authors take cells from a patient they sorted, and see if any clones increase in frequency with colony assays? In vivo studies would be of even more value but are clearly outside scope.

Response: Thank you for this feedback. As mentioned by Reviewer 1, future work must focus on determining the function (or lack thereof) of these clonal mutations *in vivo*.

Reviewer #3 (Remarks to the Author): Expert in haematopoiesis and sequencing

This manuscript describes sensitive detection of clonal hematopoiesis in paired blood specimens, banked ~10 years apart, from 20 healthy individuals aged medians of 56.6 and 68.1 years old, using barcoded targeted capture sequencing of 54 AML-associated genes. The most significant finding here is very high frequency of clonal hematopoiesis, which tended to be stable over years and nevertheless, increase with the second samplings; 19 of 20 had subclones carrying 1-17 somatic mutations, which frequently involved epigenetic regulators, such as *DNMT3A* and *TET2*. At the second time point, 13.8% of the mutations found at the first time point were lost, whereas 27.5% persisted and 58.7% were newly acquired. The results are of potential interest, but there are several concerns to be addressed.

Major points:

1) The significance of the present study critically depends on the accuracy of detection of mutations having very low allele frequencies. Authors evaluated the accuracy by seeing distribution of mutations in terms of coverage per amplicon and the number of mutations per gene, as well as ddPCR for selected mutations. My concern is that ddPCR validation is biased for mutations having high prior probability, such as those in *DNMT3A* and *TET2*. How is the accuracy of mutations in other less frequently mutated genes having low allele frequency?

Response: While *DNMT3A* and *TET2* were the most commonly mutated genes, there is no *a priori* reason why mutations in those genes would be more likely to validate by an orthogonal method (ddPCR) as mutations of similarly low frequency in other genes. As shown in Supplementary Table 4, we also used ddPCR to validate mutations found in *STAG2*, *TP53*, *BCORL1*, *CBL* and *KRAS*. Our validation rate was 100% with high correlation between the VAFs detected by ECS and ddPCR. These variants were detected across multiple samples at frequencies similar to the validated *DNMT3A* and *TET2* mutations. Additionally, we detected and quantified the VAF of 11 mutations using ddPCR in the first time point, which were only called by ECS in the second time point (see the dashes under "Collection 1" for Participants 5, 7, 9, 11, 12, 13, 18, 19). Together, these findings demonstrate that, if anything, our ECS variant calling strategy lacks sensitivity in favor of high specificity. Only one variant that was not detected by ECS in the first time point, *TET2* S1486X in individual #9, was also not detected by ddPCR at that time point. Given the concerns over false positives raised by Reviewer 1 and explained in our response above, we have tuned the bioinformatics to accept false negatives, as elucidated by the ddPCR validation experiments, in order to reduce the number of false positives.

Additionally, to demonstrate that the ddPCR results for rare events was not artifactual, we compared a negative control sample from the NHS cohort against each test sample assayed by ddPCR (see the last two columns in Supplementary Table 4). The control sample was assayed with the same number of wells and droplets as the test sample. In every case, there were so few positive droplets in the control sample that we could determine our expected test mutation was significantly different from the negative control. These results demonstrate that ddPCR is a very sensitive and

specific tool for validation and that calls made by ECS are very specific. Unfortunately, ddPCR cannot be used for discovery, but is excellent for validation.

2) There was discordance in the results of the duplicated experiments for several mutations within the list of 109 mutations (Supplementary Table 2); some mutations were not detected or showed substantially different VAFs in duplicated samples. These mutations should not be considered as true positive if ever the duplicated experiments are meaningful. Otherwise, all such samples should be intensively validated by ddPCR or other independent experiments.

Response: This is an excellent point, which highlights the necessity of using replicates for error-corrected sequencing. While we reported mutations detected in a single replicate in the supplementary tables as a matter of transparency, the primary figures only contain data where the same mutation was observed independently in both replicates at a given time point. However, we have gone back to manually review the raw data from mutations that were only detected in a single replicate. Often, we find evidence that the variant was present in the both replicates, but did not have enough coverage or failed one of the quality thresholds in the variant calling software (i.e. coverage <1000x, variant coverage <5x, Bonferroni-corrected binomial p-value>0.05). Again, we tolerate these false negatives to improve the specificity of our variant calls. Since variant calling from ECS data is a new technique, we spent a large amount of time testing the parameters of the variant calling. In the end, we decided to lean towards more stringent variant calling such that we would under-report the prevalence of clonal hematopoiesis, but be confident that variants we did call would validate by ddPCR, regardless of frequency or gene.

3) In Supplementary Figure 4, which of the results from duplicated experiments was used for the plots, or authors used mean values? Again, some results in duplicated experiments were very different. Also it does not seem to be appropriate to analyze very different VAFs (ranging from 0.0003-0.1451 (~500 times!)) in the manner presented. The plot should be in log scale and statistical method should be improved.

Response: This is a good point that a linear scale is not appropriate given the range of VAFs validated. Figure 4 was replotted with a log scale and we hope the reviewer will find this presentation of the data clearer. This plot includes the VAF measured independently from each replicate (not their mean) compared to the VAF measured by ddPCR for that sample.

4) Authors are requested to test whether or not newly detected mutations in the second time point was present from since the first time point using ddPCR.

Response: In supplementary table 4, we provide results from the requested experiment for 11 variants that were detected by ECS in only the second time point (see my response to point #1 above). Of the 11 variants not called by ECS in the first time point, 10 were detected by ddPCR. Again, we believe this demonstrates the high specificity of the ECS method.

5) Were there any other mutations showing >0.2 of VAFs?

Response: There was a single mutation detected at approximately 0.35 VAF that likely occurred in an individual where heterozygous variant occurred in a region with a segmental duplication of the wild-type allele. There were numerous, presumably heterozygous, mutations that were detected at approximately 0.5 VAF.

6) Authors speculated that multiple *DNMT3A* and *TET2* mutations may not resided in the same clone. It should be tested whether or not these mutations were found in the same sequencing reads for those mutations located in the vicinity, if ever present.

Response: This is another excellent point. When we examined the raw error-corrected sequencing reads we did find a few cases where mutations co-occurred in the same reads and, likely, originated in the same cell. We have added these data points into the discussion section. However, we did not want to overstate our ability to co-localize mutations, which in our eyes is the top limitation of the methodology. Regardless, we changed/added the following text to the discussion section:

“Additionally, we could not routinely co-localize mutations within the same hematopoietic clone. However, we co-localized variants in three instances where variants co-occurred in the same sequenced reads (PID 2, *TET2* R1216G/A1217A; PID 13, *DNMT3A* G498V/C494F; PID 14, *KRAS* A66A/S65S).”

7) Authors generated a nucleotide position-specific error profile for each sample, using all sequenced libraries that were not from the same individual. But this would lead to elimination of hotspot mutations that may be present recurrently in different individuals, such as *RAS* mutations. There was low frequency of R822 *DNMT3A* mutations and other hot spot mutations.

Response: This is an astute observation made by the reviewer. We also had this concern. While preparing this manuscript, we selected the hot spot mutations used for the analysis in the McKerrel et al (*Cell Reports*, 2015) manuscript to look in the raw data for evidence of missed hot spot mutations. To our surprise, of the hot spots that were covered by our panel, we only found one case of a likely missed mutation that was omitted by our analysis (a *DNMT3A* R882H mutation found in individual #5). This mutation was omitted because it occurred at a lower frequency (approximately 0.0005-0.001 VAF) than the R882H mutation identified in individual #13 at 0.002 VAF. We added the following text to the discussion section to highlight this limitation of the method:

“Additionally, the binomial statistical framework will underreport hotspot mutations occurring in multiple individuals. However, in our raw data we only observe a single likely instance of an uncalled hotspot mutation—a *DNMT3A* R882H variant in individual 5 observed at a lower frequency than the variant reported in individual 13.”

8) Many intronic SNVs were found in *DNMT3A* and other genes (Supplementary Figure 8), which is also of interest. Given that these mutations are likely non-functioning, what was the status of mutations in other AML-related genes to explain what mutations or genetic events drove the clonal evolution of cell having non-functional mutations. Also were there any specific mutation signatures, such as transcription-coupled repair?

Response: This is an interesting point that we have thought about. Ultimately, we thought our explanation for selection of intronic mutations was too speculative to include in the text. Many of the intronic mutations overlap with transcription factor binding sites, which if disrupted may lead to aberrant expression of the underlying gene. These are most evident in the indels that we identified, where 2/3 intronic indels overlapped with putative binding sites for transcription factors related to

hematopoiesis. We would be pleased to include a comment on this in the revised text if the reviewer and editor feel it is appropriate.

Chr	Start	End	Ref	Alt	Gene	TF Binding Site
2	25463381	25463381	-	GTG	DNMT3A	None
2	25467528	25467547	AGCAGCGGGAAGGGTCAGAA	-	DNMT3A	CTCF
X	123179310	123179318	ATTAATTTT	-	STAG2	STAT3, CEBPB

Reviewer #1 (Remarks to the Author):

The authors have done an excellent job answering the reviewer questions.

Reviewer #3 (Remarks to the Author):

Authors satisfactorily addressed most of the criticisms and questions this reviewer raised. I have only one concern, which I think was not fully answered.

Response: This is an excellent point, which highlights the necessity of using replicates for error-corrected sequencing.

"While we reported mutations detected in a single replicate in the supplementary tables as a matter of transparency, the primary figures only contain data where the same mutation was observed independently in both replicates at a given time point."

So the 109 SNVs are not included in the primary figures, even though the main text said "We identified 109 clonal single nucleotide variants (SNVs) below 0.2 VAF in 95% (19/20) of individuals"? Authors stated in the rebuttal "In the end, we decided to lean towards more stringent variant calling such that we would under-report the prevalence of clonal hematopoiesis", but they actually called all the 109 variants as true positive. Very confusing and misleading. Authors should make clear in the main text how they used the results in the duplicated samples to call positive variants. Otherwise, readers misunderstand that these 109 variants are positive in both of the duplicated samples and how they considered positivity only in a single sample.

Otherwise, I have no further concerns.

Reviewer #1 (Remarks to the Author):

The authors have done an excellent job answering the reviewer questions.

Reviewer #3 (Remarks to the Author):

Authors satisfactorily addressed most of the criticisms and questions this reviewer raised. I have only one concern, which I think was not fully answered.

Response: This is an excellent point, which highlights the necessity of using replicates for error-corrected sequencing. "While we reported mutations detected in a single replicate in the supplementary tables as a matter of transparency, the primary figures only contain data where the same mutation was observed independently in both replicates at a given time point."

So the 109 SNVs are not included in the primary figures, even though the main text said "We identified 109 clonal single nucleotide variants (SNVs) below 0.2 VAF in 95% (19/20) of

individuals"? Authors stated in the rebuttal "In the end, we decided to lean towards more stringent variant calling such that we would under-report the prevalence of clonal hematopoiesis", but they actually called all the 109 variants as true positive. Very confusing and misleading. Authors should make clear in the main text how they used the results in the duplicated samples to call positive variants. Otherwise, readers misunderstand that these 109 variants are positive in both of the duplicated samples and how they considered positivity only in a single sample. Otherwise, I have no further concerns.

Response: Thank you for highlighting this potential confusion. It is not our intention to mislead the audience. Perhaps the confusion lies in the distinction between two independent technical replicates per time point and two time points per individual (there are four independent sequencing libraries for each individual). Mutations were called independently in each sequencing library. A mutation was only reported as positive at a given time point if it was called independently in both replicates for that time point. In some cases, a mutation was only called at a single time point, but this means it was observed in both replicates at that time point. When a mutation was called at both time points, it was observed in all four sequencing libraries. In some cases, a mutation was called in both replicates at a single time point, but only one replicate in the second time point – in this case, the mutation was not called at that second time point even though it may be a true positive, suggesting that we are potentially undercalling mutations. This is supported by our ddPCR validation experiments. Because this is a central tenet of the manuscript, we take this seriously and have attempted to make this clearer in the Results and Methods sections.

The method section already describes the variant calling criteria: "A variant was called if the binomial p-value was: a) less than 0.05 after Bonferroni correction, b) the variant was observed in at least 5 ECCSs, c) the VAF was greater than 0.0001, and d) the variant was identified with criteria a-c in at least two replicates from one of the two time points."

The 109 SNVs reported were the variants identified independently in both replicates from at least one time point. We did not intend to suggest that all 109 variants were detected at both time points, and these differences should be clear in Supplementary Table 2. However, to eliminate any confusion we have reworded that specific statement in the Results section to clearly indicate that the 109 SNVs were detected in at least one time point (and not both): "We identified 109 clonal single nucleotide variants (SNVs) in at least one time point below 0.2 VAF in 95% (19/20) of individuals."

Additionally, in the Results section, we indicated how many variants were detected at each time point: "Of the 109 clonal SNVs identified, 27.5% (30/109) were detected at both time points, 13.8% (15/109) were detected at only the first time point, and 58.7% (64/109) were detected at only the second time point (Supplementary Table 2)." In Supplementary Table 2, we listed the VAF detected in each technical replicate even if both replicates from a given time point did not identify the variant (i.e. the 3 out of 4 libraries example mentioned above). Since these were not included as true positives *at that time point*, if you removed any mutation identified in only a single technical replicate from Supplementary Table 2, there would still be the 109 SNVs detected in this study. We apologize for the confusion and hope this clarifies our analysis.

Thank you for your feedback regarding this manuscript.